# Oxidised Low-Density Lipoprotein-Induced Platelet Hyperactivity—Receptors and Signalling Mechanisms

**DOI:** 10.3390/ijms23169199

**Published:** 2022-08-16

**Authors:** Martin Berger, Khalid M. Naseem

**Affiliations:** 1Department of Internal Medicine 1, University Hospital Aachen, Pauwelstraße 30, 52074 Aachen, Germany; 2Leeds Institute of Cardiovascular and Metabolic Medicine, The LIGHT Laboratories, Clarendon Way, University of Leeds, Leeds LS2 9JT, UK

**Keywords:** oxidised LDL, oxPC_CD36_, CD36, lectin-like oxidised LDL receptor-1, scavenger receptor A, toll-like receptor, myocardial infarction

## Abstract

Dyslipidaemia leads to proatherogenic oxidative lipid stress that promotes vascular inflammation and thrombosis, the pathologies that underpin myocardial infarction, stroke, and deep vein thrombosis. These prothrombotic states are driven, at least in part, by platelet hyperactivity, and they are concurrent with the appearancxe of oxidatively modified low-density lipoproteins (LDL) in the circulation. Modified LDL are heterogenous in nature but, in a general sense, constitute a prototype circulating transporter for a plethora of oxidised lipid epitopes that act as danger-associated molecular patterns. It is well-established that oxidatively modified LDL promote platelet activation and arterial thrombosis through a number of constitutively expressed scavenger receptors, which transduce atherogenic lipid stress to a complex array of proactivatory signalling pathways in the platelets. Stimulation of these signalling events underlie the ability of modified LDL to induce platelet activation and blunt platelet inhibitory pathways, as well as promote platelet-mediated coagulation. Accumulating evidence from patients at risk of arterial thrombosis and experimental animal models of disease suggest that oxidised LDL represents a tangible link between the dyslipidaemic environment and increased platelet activation. The aim of this review is to summarise recent advances in our understanding of the pro-thrombotic signalling events induced in platelets by modified LDL ligation, describe the contribution of individual platelet scavenger receptors, and highlight potential future challenges of targeting these pathways.

## 1. Introduction

Thrombotic events associated with atherosclerotic plaque rupture and unwanted platelet activation account for almost 25% of deaths worldwide and represent a major burden on the health of most countries [1]. Dyslipidaemia, secondary to conditions such as type 2 diabetes, obesity, chronic kidney disease, and metabolic syndrome, is associated with both atherogenesis and thrombosis [2]. Studies in mice and humans strongly suggest that dyslipidaemia induces a prothrombotic phenotype, in which platelet hyperactivity plays a major role [3,4,5,6,7,8]. Indeed, the increased thrombotic risk associated with dyslipidaemia may contribute equally to its atherogenic effects the predisposition of disease [9]. A biochemical characteristic of dyslipidaemia is increased oxidative stress and the accumulation of oxidised lipids in the vessel wall, in the form of oxidised low-density lipoproteins (oxLDL), as well as in the plasma [10]. These particles play an important role in promoting atherogenesis through endothelial cell activation, the proliferation of vascular smooth muscle cells, and the formation of lipid-laden foam cells [11]. However, there is compelling evidence to indicate that these modified forms of the lipoproteins also exist in the circulation, particularly in subjects at risk of atherothromboic disease [12,13,14,15,16]. In humans, the presence of circulating oxidatively modified LDL was first reported by Avogaro and colleagues, who described a modified sub-fraction of LDL with increased electrophoretic mobility on ion exchange chromatography and elevated levels of lipid peroxidation-induced conjugated dienes [17]. The presence of circulating oxLDL was given further credence by studies that identified the autoantibodies of oxLDL in subjects with coronary artery disease (CAD) and animal disease models [18,19,20,21,22]. Autoantibodies of oxLDL isolated from ApoE-deficient hyperlipidaemic mice bind exclusively to oxidised phospholipids [19] and lead to the development of several monoclonal antibodies against oxidatively-modified epitopes on LDL, thus allowing for the determination of concentrations of oxLDL plasma of human subjects [12,23,24,25,26]. The presence of elevated levels of modified LDL has been observed in end-stage renal disease, obesity, hypercholesterolemia, diabetes, and coronary artery disease (CAD) [27,28,29,30,31]. More recently, a similar electronegative LDL representing the fifth quintile of the electrophoretically separated LDL (L5), with similar biochemical properties to LDL that had been oxidised in vitro, was identified in subjects with both CAD and stroke [32,33]; it was critically shown to be biologically active ex vivo. Moreover, in subjects with acute coronary syndrome (ACS), the platelets circulate with oxLDL bound to their surface, suggesting a functional relationship in the disease environment [34,35].

The link between dyslipidaemia and elevated platelet activation was first described by Carvalho in subjects with familial hypercholesterolaemia [8]. These critical findings were confirmed in animal models of hyperlipidaemia and shown to impact directly on thrombosis [7], with even mild hyperlipidaemia leading to accelerated platelet activation [4]. The levels of oxLDL in the circulation predicted future atherothrombotic events and the presence and severity of atherosclerotic plaque [12,31,36]. Given this context, a significant number of investigators have shown that oxLDL can both induce and promote the activity of platelets when tested in vitro (Table 1). Thus, there is strong evidence to suggest that oxidised LDL may be a primary causative factor in inducing platelet hyperactivity and the associated prothrombotic phenotype observed in a number of disease cohorts [37,38,39], both in the acute setting of plaque rupture and chronically while the platelets are in circulation. In the current article, we review the biochemical mechanisms that translate atherogenic lipid stress, in the form of oxLDL from the circulation, into platelet hyperactivity.

## 2. Oxidised LDL Are Heterogeneous Particles

LDL are circulating lipid-rich particles whose primary function is the delivery of cholesterol, in the form of cholesteryl esters and fat-soluble vitamins, to non-hepatic cells. They are composed of a hydrophobic core of cholesteryl esters and triglycerides, surrounded by a monolayer shell of phospholipids. Native LDL also contain a single polypeptide chain, apolipoprotein B_100_, which is required by the ApoB/E receptors on the cell surface for recognition and endocytosis. It is important to recognise that LDL represent a heterogenous group of particles whose biochemical composition is determined by diet, genetics, and physical activity [98]. This biochemical heterogeneity occurs within individuals and between individuals with any given population. The composition of the particles also has a direct influence on both their resistance to oxidation and the types of biochemical lipid species that are produced in response to oxidative attack. The oxidation of LDL in vivo is thought to occur in the subendothelial space, when the particles can become anchored by glycosaminoglycans and are subsequently exposed to a number of factors that induce oxidative modifications, including reactive oxygen species, reactive nitrogen species, and cellular enzymes [98]. In vitro this can be mimicked by exposing isolated LDL to a range oxidizing systems, including cells, free radical generating systems, non-radical oxidants, and enzymes [98,99]. While, in this review, we focus on LDL that has been altered by oxidation, it is important to note that other forms of modification have been reported including glycation (diabetes mellitus), carbamylation (chronic kidney disease), and acetylation, all of which may contribute to platelet activation in different circumstances [27,30,74,100].

The term oxidised LDL is often used as a “catch all” term used to describe LDL preparations, either isolated directly from the circulation or, more commonly, preparations that have been oxidised under very defined experimental conditions. Given the complex and varied biochemical composition of LDL particles, a vast array of bioactive compounds have been reported to be present, including modified phospholipids, sphingolipids, free fatty acids, cholesterol esters, oxysterols, and aldehydes [99]. The generation of various combinations of these species creates a complex population of particles, with potentially distinct biological properties. Oxidation is a progressive event, beginning with the loss of endogenous antioxidants, including β-carotene and α-tocopherol, progressing to lipid oxidation, and finally, the modification of the apoB_100_, as measured by ELISA or mass-spectrometry [101,102]. From an experimental perspective, oxLDL preparations are generally divided into two types, minimally modified LDL (mmLDL or mildly oxidised LDL) and oxLDL (fully oxidised). MmLDL represents particles that have undergone lipid peroxidation, but not protein modification. The process begins with the peroxidation of the fatty acid chains of polyunsaturated fatty acids of the particle shell before progressing to cholesteryl esters in the core. These mmLDL particles can still be recognised by the apoB/E receptor and could, therefore, exist in circulation for extensive periods [103]. In contrast, oxLDL have undergone extensive lipid oxidation coupled to protein modification and most likely represents the type of particles present in atherosclerotic plaques that may be released upon rupture. Here, the continued oxidation of the lipids leads to the generation of aldehydes, such as malondialdehyde and 4-hydroxynonenal, which form adducts with the lysine residues of the ApoB_100_, thus resulting a net loss of the overall positive charge [99,101]. These progressive biochemical changes diminish the recognition of LDL by the apoB/E receptor and lead to the increased recognition by a series of scavenger receptors. Given the continual generation of LDL in the blood stream, as part of the endogenous lipid metabolism pathway, atherosclerotic plaques probably contain a spectrum of modified LDL species, although it is unclear if the same is true in the circulation. Therefore, it is critically important to appreciate that, when assessing the effects of modified LDL on platelets, regardless of their origin or method of oxidation, the cells are being exposed to a diverse population of particles with non-uniform biochemical composition and properties. The difference in bioactive compounds produced by these varied approaches probably account for the plethora of effects these particles have been shown to induce in platelets.

## 3. The Role of oxLDL in the Pathogenesis of Metabolic Disease 

The presence of oxLDL is linked to several disease states, and elevated levels of oxLDL in the plasma have been detected in a varity of disorders, including coronary artery disease, dyslipidaemia, stroke, acute coronary syndrome, metabolic syndrome, chronic kidney disease, and several autoimmune diseases (Figure 1) [14,15,28,29,104]. Interestingly, the many lines of evidence suggest that oxLDL has a causal relationship with the pathogenesis of atherosclerosis, diabetes, and certain autoimmune disease (reviewed in [98]). This causal role is particularly well-established in the context of atherosclerosis, and oxLDL has been found to affect the early stages of the disease, including the endothelial dysfunction by endothelial cell network formation, dysregulation of endothelial nitric oxide production, and alignment of endothelial cells under flow conditions [105,106,107]. Later, the accumulation of the modified cholesterol species, derived from oxLDL in macrophages and leading to foam cell formation, in the intima is critical to driving proinflammatory processes in the vessel wall [108,109]. The causality is underpinned by three observations—(i) oxLDL is only found in diseased arteries [110], (ii) downregulation of scavenger receptors that serve as ligands for oxLDL greatly reduce atherosclerosis disease burden in mice [111], and (iii) in Apo-E deficient mice presence of oxLDL precedes the development of an atherosclerotic plaque [112]. In patients with diabetes, the glycaemic environment increases the susceptibility of LDL to oxidation and, therefore, increases its atherogenic potential [113]. In addition, the presence of oxLDL leads to an impairment of glucose transporter 4 (GLUT4) and desensitises adipocytes against insulin, therefore increasing the hyperglycaemic state [113,114]. The notion that inflammation promotes LDL oxidation has led many researchers to investigate the role of oxLDL in various autoimmune disease. Even though the causal link in pathogenesis is less clear in these patients, the presence of oxidised LDL or anti-oxLDL immunocomplexes is linked to disease severity and progression in Rheumatoid arthritis, Systemic Lupus erythematodes, Sjogrens’s syndrome, and several vasculitides [115,116,117]. In summary, oxLDL has been linked to many disease states and, therefore, serves as an interesting biomarker; however, and while uncertainties still exist, evidence suggests a causal link, particularly in atherosclerosis, which conceptually makes oxLDL an interesting therapeutic target.

## 4. Functional Responses of Platelets to Modified Low-Density Lipoproteins

The ability of oxidatively modified forms of LDL to activate and sensitise platelets for activation has been appreciated for over forty years; since that time, our knowledge of these effects and the associated mechanisms have increased dramatically. The study by Avogaro, identifying oxidised LDL in the plasma of patient with atherosclerosis [17], suggested that these modified lipoproteins may be the link between dysregulated lipid metabolism and platelet hyperactivity. A large number of studies have been performed examining platelet function in response to various forms of laboratory oxidised LDL. These atherogenic lipoproteins activate a plethora of signalling events in the platelets that underpin their ability to augment platelet functions. Examination of the literature demonstrate that oxLDL can influence many critical platelet functions, including secretion [34,56,57,58,59], adhesion [34,40,41,42,43,44,45,46,47,48,49,50,51,52,53,54], microvesicle release and pro-coagulant phosphatidylserine expression [96], Ca^2+^ mobilisation [63,68,83], reactive oxygen species generation [42,89,90], and act synergistically with physiological agonists to promote secretion and aggregation [33,63,68,79,80,81,118] (Table 1). 

Furthermore, immobilised oxLDL supports platelet adhesion and spreading, both as independent ligands and through the potentiation of the adhesive capacity of extracellular matrix proteins [44,89]. It is noteworthy that there is robust evidence suggesting that the exposure of platelets to mmLDL and oxLDL primes the cells for activation and potentiates the effects of haemostatic agonists; the evidence supporting oxLDLs as agonists capable of induce aggregation is inconsistent. The reasons for this are unclear, but likely reflects the use of distinct LDL preparations that have a specific combination of lipid oxidation products.

The two major considerations when evaluating the potential for oxLDL augmented platelet functions are the platelet model used and type of LDL preparation. The vast majority of studies highlighted in Table 1 use washed platelets and occasionally platelet-rich plasma (PRP), but not whole blood. Experiments with washed platelets allow for the investigation of lipoproteins in a controlled environment and represent important proof-of-principle; however, it is clear that platelet phenotypes differ according to the presence of buffer, plasma, or whole blood (Table 1). The effects of oxLDL are clearly most potent when incubated with washed platelets and less so with PRP. It possible that the antioxidant environment of the plasma protects agonist potent oxidised lipids, thus diminishing their activatory capacity. Nevertheless, the oxLDL has been shown to be effective in whole blood, where it can act as a mild agonist that influences platelet adhesion, secretion, and thrombosis [79,88]. Therefore, LDL are likely to have a major impact at sites of atherosclerotic plaque rupture, where platelets would be exposed to both various forms of modified LDL and a plethora of physiological platelet activators. While this has not been demonstrated directly, a recent study demonstrated that the infusion of oxLDL into mice significantly enhanced thrombosis using a ferric chloride model [88]. Given that oxLDL may induce procoagulant effects on the endothelium [119,120], these experiments need to be interpreted with caution; nevertheless, they are an important proof of principle that oxLDL promotes thrombosis in vivo.

The second important consideration is the method and extent of oxidation. It has been proposed that mildly oxidised LDL are more proficient in driving platelet activation, and they may have more functional relevance, given that they could potentially exist for extended times in circulation. Korporaal and colleagues compared several degrees of LDL oxidation, ranging from 0–60%, based on the number of conjugated dienes, and found that 60% lipid peroxidation (likely classified as mmLDL) is required to achieve a significant aggregatory response [94]. Similarly, both Weideman et al. and Naseem et al. showed that the same LDL preparation, oxidised to different extents, produced distinct functional responses with the mildly oxidised inducing aggregation, which would fully oxidise LDL, with little or no effect [73,75]. This picture is further complicated by the type of oxidation, with early studies suggesting that hypochlorous acid modified LDL was a more potent platelet activator than Cu_2_SO_4_-induced oxLDL, with LDL inducing full platelet aggregation, while copper-oxidised LDL only showed minor effects [63,74]. These studies present an extremely complex picture of the activatory effects, influenced by an array of biologically active compounds found in different quantities that are produced under these conditions. Whilst these various modified LDL preparations can present a confusing picture, with regard to their effects on platelets, it is important recognise that the heterogeneity of modified LDL closely resembles the in vivo situation, where there will be LDL at various stages of oxidation that have been exposed to multiple sources of oxidative insult. In this regard, several critically important studies have examined the clinical relevance of these studies using LDL oxidised in vitro by isolating oxLDL from patients and testing their effect on platelet function. Definitive links to in vivo modified LDL were shown much later, with Colas demonstrating that LDL taken from obese subjects with metabolic syndrome or subjects with type 2 diabetes were peroxidised and had the ability to potentiate platelet aggregation and TxA_2_ formation [28]. Two studies from Chan and colleagues isolated an electronegative LDL from patients with ACS and ischaemic stroke, which had a similar biochemical composition to in vitro copper oxidised LDL [32,33]. In vivo studies demonstrated that this modified electronegative faction of LDL could potentiate platelet activation, but not induce aggregation. These important observations justify the enormous efforts to identify the mechanisms of oxLDL-induced platelet activation and suggest that targeting this type of platelet activation may provide context-dependent platelet inhibition, without compromising key haemostatic pathways.

## 5. Scavenger Receptors—The Transducers of oxLDL into Platelet Hyperactivity

The cellular recognition of modified LDL was initially described in the landmark study by Goldstein and colleagues, who demonstrated the ability of macrophages to recognise and engulf acetylated LDL, independently of the apoB/E receptor [109]. The binding of these modified lipid bearing particles is facilitated by a number of different scavenger receptors, of which, there are five different classes with the capacity to bind to a plethora of danger- and pathogen-associated molecular patterns (DAMPs, PAMPs) [121]. The binding of modified LDL to platelets has been demonstrated by a variety of techniques, including radioactive labelling, flow cytometry, and fluorescence microscopy [74,92,122]. Ligation has been proposed to occur through several platelet scavenger receptors, including cluster of differentiation 36 (CD36), lectin-like oxidised low-density lipoprotein receptor-1 (LOX-1), scavenger receptor B1 (SR-B1), G-coupled receptor lysophosphatidic acid receptor 1 (LPAR1), and integrins [123]. The recognition of oxLDL by the scavenger receptors is dependent on specific lipid oxidation products, their relative amounts, and the modification of the apoB_100_ by lipid derivatives. These receptors show distinct preferences for particular forms of oxidatively modified LDL; although, given the complex and progressive nature of the biochemical modification, there is a likely overlap, where modified LDL may interact with more than one type of scavenger receptor (Figure 1). The presence of multiple scavenger receptors as binding sites for oxLDL, combined with the heterogenous composition of the particles and differences in oxidation methods, underpins the varied responses to these lipoproteins observed in platelets. However, despite this complexity, the ligation of all of scavenger receptors by modified LDL is generally linked to an activatory signalling in platelets. Understanding the contribution of these individual receptors to the prothrombotic phenotype in disease states will allow for a better understanding of how and why oxLDL-induced signalling promotes unwanted platelet activation.

### 5.1. CD36—Class B Scavenger Receptor

CD36 is a heavily glycosylated transmembrane receptor consisting of two short intracellular segments at the amino and carboxy termini, two transmembrane domains, and a large extracellular loop [124,125]. CD36 is highly expressed on the platelet surface, with up to 20,000 copies per platelet [126,127]. It was first identified as GPIV as a receptor for collagen and then later found to have a high affinity for plasma protein thrombospondin-1 [128,129]. The identification of CD36 as a receptor for oxLDL was reported by Endemann et al., who demonstrated that CD36-transfected HEK cells bind oxLDL [130]. CD36 does not recognise delipidated LDL, thus indicating that the ligand recognition moieties within oxLDL are primarily oxidised phospholipids [131]. This was confirmed by Terpstra and colleagues, who showed that the recognition by CD36 was mediated by oxidised lipids present in oxLDL, which appeared early in the oxidative process, and were likely ligands for CD36, since these were effective antagonists of oxLDL binding to macrophage CD36 [132]. A critical breakthrough in our understanding of CD36 ligation was the systematic investigation of phospholipids present in oxLDL, as reported by Podrez and colleagues. This study demonstrated that oxidation of the two most abundant molecular species, i.e., choline glycerophospholipids 1-palmitoyl-2-arachidonyl-sn-glycero-3-phosphocholine (PAPC) and 1-palmitoyl-2-linoleoyl-sn-glycero-3-phosphocholine (PLPC), lead to generation of several highly CD36-specific phospholipids [133]. These phospholipids have been shown to be dramatically increased in the plasma of genetic models of hyperlipidaemia, including both apoE and LDLr deficient mice. Importantly, it was recognised that only one or two of the oxidised molecules needed to be present in the LDL to enable recognition and binding, thus reiterating that only minor changes in the oxidative modification process are required to alter platelet function. Elegant translational studies from Podrez and colleagues build on these observations to demonstrate the key link between oxidative lipid stress, CD36, and platelet hyperactivity. Using genetically modified mice hyperlipidemic mice, the authors demonstrated the presence of increased plasma levels in these oxidised phospholipids, which was associated with a platelet-driven prothrombotic phenotype and corrected by the deletion of CD36 [61].

OxLDL binds to CD36 in a region located between amino acids 155–183, with a particular requirement of the positively charged, highly conserved amino acids lysine164 and lysine166 [134,135]. The ligation of CD36 by oxLDL or oxidised phospholipids leads to the activation of a complex and interlinked signalling pathway that is linked to multiple aspects of platelet function. Critical to the ability to transduce oxLDL binding to the intracellular signalling machinery of the platelets are the Src family kinases (SFK), two of which, Fyn and Lyn, are constitutively associated with platelet CD36 [136]. We, and others, have shown that ligation of oxLDL to CD36 leads to activation of a tyrosine kinase signalling pathway that is initiated by the activation of SKF [44,46,59], which links CD36 to multiple downstream signalling pathways (Figure 1). The use of SFK kinase inhibitors indicates that Fyn and/or Lyn are functionally linked to ERK, p38, Mkk4/JNK, Vav1/3, and Syk [46,52,59,90,137]. Mice that are deficient in Vav1 and Vav3 are protected against the prothrombotic phenotype displayed in ApoE^−/−^ mice, suggesting a critical role for this protein in promoting atherothrombosis [137]. The establishment of a role of serine/threonine kinases of mitogen-activated protein kinases (MAPK) in oxLDL-mediated macrophage foam cell-formation led Chen et al. to describe the sequential activation of Src, MKK-4, and JNK in response to oxLDL, thus leading to activation of a_IIb_b_3_ and P-selectin expression [59]. Korporaal and colleagues linked oxLDL activatory signalling to the MAPK p38, but the necessity of SFKs was not investigated [52]. Consistent with these studies, Karimi et al. showed that inhibition of p38 and JNK lead to a reduced functional response to oxLDL treatment [138]. Most recently, SFK signalling has been associated with increased ROS generation and subsequent Caspase-3 cleavage, which lead to increased phosphatidylserine expression and might link oxLDL ligation and SFK signalling to pro-coagulant activity of platelets [139].

Consistent with canonical signalling from GPVI, SFK couple CD36 to the activation of Syk, which leads to the activation of a RhoA-dependent inhibition of myosin light chain phosphatase that underpins spreading and shape change [46]. Downstream signalling from Syk leads to phosphorylation of phospholipase Cγ2 (PLCγ2), which appears to be critical for oxLDL-induced platelet activation, since either pharmacological inhibition or genetic deletion blunted platelet hyperactivity [46,89,140]. The role of Syk in activating the other key signalling elements, such as ERK, p38, Mkk4/JNK, and Vav1/3, is unclear; although, it is likely, given that Syk activation lies downstream of other tyrosine kinase-linked receptors in platelets, for example, GPIb-IX-V, GPVI, Clec-2, and α_IIb_β_3_ [141,142,143,144]. The sequential activation of Syk and PLCg2 leads to the activation of protein kinase C (PKC) isoforms. While it remains unclear which PKC isoforms lie downstream of CD36, their activation is critical to its ability to modulate cyclic nucleotide signalling in platelets. PKC-mediated activation of NADPH oxidase 2 (NOX2)-dependent ROS production inhibits PKG signalling [140], while activation of the phosphorylation and activation of PDE3A modulates of PKA signalling (Figure 1) [88]. This process has been termed disinhibition and leads to a reduced threshold for platelet activation and provides the molecular mechanisms for earlier observations [87]. Thus, a substantial body of work suggests that oxLDL ligation to CD36 promotes platelet activation, blunts inhibitory signalling, and contributes to the generation of procoagulant platelets; the combination of these interrelated pathways promotes the potential for platelet hyperactivity and contribute to dyslipidaemia-induced procoagulant state.

The role of CD36 as a critical receptor for platelet hyperactivity, in the context of lipid imbalances, has gained further importance with two very recent studies. Proprotein convertase subtilisin/kexin type 9 (PCSK9), which can regulate cholesterol uptake in the liver, is elevated in subjects at risk of cardiovascular events and associated with increased platelet reactivity [145].

### 5.2. Lectin-like Oxidised Low-Density Receptor 1—Class E Scavenger Receptor

Lectin-like oxidised low-density receptor 1 belongs to the C-type lectin family and consists of four domains, i.e., a short N-terminal cytoplasmic domain, single transmembrane domain, connecting neck domain, and lectin-like domain at the C-terminus [146]. LOX-1 was initially identified as a receptor for oxLDL in endothelial cells [147]. LOX-1 differs in its recognition of oxLDL from CD36, since it can bind to both mildly oxidised and delipidated oxLDL, suggesting it has the capacity to recognise oxidised phospholipids and modified apoB_100_. In platelets, LOX-1 is found in the alpha-granules and relocated to the cell surface upon platelet activation [147]. This suggests that LOX-1 may facilitate the effects of oxLDL secondary to initial platelet activation and makes understanding the signalling pathways that follow LOX-1 ligation difficult to investigate, since any preactivation by alternative agonists could contribute directly to the signalling outcomes. Nevertheless, LOX-1 has been shown to play a role in platelets, in response to chromatographically separated LDL from ST-elevation myocardial infarction (STEMI) and stroke patients [32,33]. It is worth noting that electronegative LDL resembles the bioactive properties of mildly oxidised LDL and was characterised by increased levels of thiobarbituric acid-reactive substances (TBARS), a measure of lipid oxidation. This highly electronegative fraction potentiated the activatory effects of ADP or amyloid β (Aβ), in order to increase platelet aggregation, P-selectin expression, and integrin expression. Mechanistically, L5-LDL-mediated activation of platelet was linked to a signalling pathway involving PI3-kinase and AKT [32,33]. In a separate study, L5-LDL isolated from stroke patients synergised with Aβ to promote platelet activation through an IKK2, IκBα, p65, and c-Jun N-terminal kinase 1 [32]. Given that Aβ is a ligand for CD36 [148], it is possible that this pathway is activatory in response to cooperative signalling between these receptors. In contrast to other scavenger receptors, the role of oxLDL-LOX-1 in platelets has been exclusively studied, in the context of severe atherothrombotic events, including myocardial infarction and stroke. These patients have elevated levels of platelet LOX-1 expression, which may suggest the presence of preactivated platelets in these conditions. This is consistent with many in vitro studies, which demonstrated that oxLDL potentiates the activation of preactivated platelets and could suggest that the minor platelet activation associated with disease may facilitate increased responsive to circulating oxLDL. However, given the absent expression of LOX-1 under resting conditions, the role of LOX-1 under stable conditions remains unclear and deserves further investigation. 

### 5.3. Scavenger Receptor A1 (SR-A)—Class A Scavenger Receptor

SR-A was identified in macrophages as a receptor capable of recognising and endocytosing oxLDL [149]. SR-A is a trimeric membrane glycoprotein with broad ligand specificity that includes oxidised lipoproteins, heat shock proteins, and bacteria. This receptor has a high affinity for fully oxidised LDL and is thought to bind to the modified apoB_100_ through recognition that involves the loss of positively charged lysine residues. Currently, definitive evidence for the expression of SR-A on platelets is lacking, since it has not been identified in proteomic studies [127], although it is present in the human platelet transcriptome [150]. A functional role for SR-A in oxLDL-mediated platelet activation has been suggested, based on a combination of pharmacological and genetic studies. The inhibition of SR-A in human platelets and double deletion of CD36 and SR-A completely inhibited p38MAPK signalling in response to oxLDL [52]. Furthermore, the inhibition of both receptors abolished platelet spreading on fibrinogen and mimicked the phenotype of pharmacological inhibition of p38MAPK. Unfortunately, knock out mice of the individual receptors were not used in these experiments. However, the fact that pharmacological inhibition of both receptors was required to prevent spreading suggests that, under some circumstances, SR-A may serve as a coreceptor of CD36.

### 5.4. LPA Receptor

Lysophosphatic acid (LPA) associated with mmLDL has been shown to activate washed platelets [53]. The presence of the three Edg family LPA receptor 1–3 has been suggested by RT-PCR [151]. However, formal identification of the receptor on the platelets by other methodologies is lacking to date. Nevertheless, Maschberger et al. implied that functional copies of the LPA receptor might be present on the platelets by showing that LPA receptor desensitization and the LPA receptor antagonist NPTyrPA blunted the activatory consequences of mildly oxidised LDL exposure [48]. However, given the lack of studies showing receptor expression and the lack of any LPA receptor knock-out animal model, these data need further confirmation.

### 5.5. Platelet Activating Factor (PAF) Receptor

Mass-spectroscopy data indicates that the PAF receptor, i.e. a G-protein-coupled receptor that binds the derivative of phosphorylcholine PAF, is likely to be expressed at low levels on platelets [127]. Chen et al. found that oxidatively modified phospholipids present in oxLDL can activate platelets in a PAF receptor-dependent manner, based on pharmacological studies with PAF receptor antagonists [68]. OxLDL and a range of oxidised phospholipids potentiated thrombin-induced Ca^2+^ mobilisation, P-selectin expression, and αIIbβ3 activation [68]. Corroboration of the potential role of the PAF receptor in platelet hyperactivity is provide by Chan and colleagues, who found that the proaggregatory effects of patient-derived electronegative LDL was blocked by the PAF receptor inhibitor ABT-491 [33]. Given the uncertainly around the expression of PAF receptors, further work in this area is required. It is possible that very low levels of expression only become functionally relevant when presented with the appropriate oxidised phospholipid species.

### 5.6. Toll-like Receptors (TLR)

TLR are highly conserved pattern recognition receptors that recognise either danger-associated (e.g., apoptotic cells) or pathogen-associated (e.g., bacterial peptides) molecular patterns (DAMPs and PAMPs). The presence of several TLR has been shown on platelets and megakaryocytes; however, until recently, their functions remained unclear [152]. Studies in macrophages demonstrates that several constituents of oxLDL, including cholesteryl esters (CE) and oxidised phospholipids (oxPL), can bind to TLRs-mediated signalling. Biswas et al. demonstrated that both TLR-2 and TLR-6 can contribute to both human and murine platelet activation induced by a oxidised phospholipids species found in oxLDL [62]. A complex picture emerges from this work, with TLR-2 contributing independently to platelet activity, while TLR-6 may utilise CD36 as a coreceptor [62]. Critically, this study did not use oxLDL; nevertheless, it proposes some interesting questions around the potentially cooperative signalling between platelet scavenger receptors. It is possible that minimally modified LDL could act as a ligand for TLRs, as evidenced in macrophages [153].

## 6. Concluding Remarks

It is now established that oxidatively modified versions of LDL circulate in subjects who are at high risk of major adverse cardiovascular events, including obesity, diabetes, stable coronary artery disease, acute coronary syndromes, and ischemic stroke. Increased circulating levels of these particles are associated with elevated risk of atherothrombotic events and linked to heightened platelet activity. Therefore, understanding the mechanisms how these pathogenic lipid particles induce maladaptive platelets phenotypes is critical to appreciating the prothrombotic phenotype in high-risk groups. Currently, mechanisms targeting scavenger receptors to control unwanted platelet activation seems unlikely, given their widespread expression; other key functional roles will need more well-designed studies to evaluate its effect. However, a key to furthering this agenda is developing a greater understanding of how these receptors function under pathological conditions. A number of key areas require further exploration

***i. Functional reprogramming of platelets in disease cohorts.*** In mice, dyslipidaemia is associated with altered haematopoiesis and increased numbers of reticulated platelets, which are thought to be more thrombogenic [154]. However, it is unclear whether this changes the expression profile of platelet scavenger receptors. Platelets from subjects with ACS express elevated levels of LOX-1 on their surface, suggesting that scavenger receptor function may be upregulated in disease [47]. Similarly, the surface expression levels of CD36 can vary by up to 50 times, which, in turn, can affect the functional response to oxLDL [126]. Understanding the expression profiles and functionality of the receptors in disease cohorts may be critical to linking oxLDL-induced platelet hyperactivity and thrombosis in disease subjects.

***ii. Receptor crosstalk.*** There are several examples in the literature that suggest cooperative partnerships between scavenger receptors, particularly CD36, which can induce many of its cellular effects through the formation of a diverse array of heteromeric signalling complexes. A range of transmembrane proteins, such as CD9, TLR2, TLR4, TLR6, CD47, and β1 integrin, can be associated with CD36 in a context-dependent manner [121], although only CD9 interacts in platelets [155]. The formation of hetromultimeric signalling complexes between different scavenger receptors or individual scavenger receptors with other adaptor proteins could be critical to the orchestration of distinct platelet responses in response to oxLDL.

***iii. Scavenger receptor-linked signalling proteins as biomarkers.*** A recent report indicated that the phosphorylation of acethyl-CoA carboxylase (ACC) in platelets, potentially downstream of CD36, was causally linked to poorer outcomes in ACS patients [156]. This is the first example of a scavenger receptor-linked biomarker for thrombotic risk and could open up the possibility of further specific early appearance biomarkers for preventive of thrombosis.

It is noteworthy that current anti-platelet medication is geared towards treating the consequences of platelet activation. A better understanding of the role of scavenger receptor may open up the possibility of targeting the cause of platelet hyperactivity in high-risk groups.

## Figures and Tables

**Figure 1 ijms-23-09199-f001:**
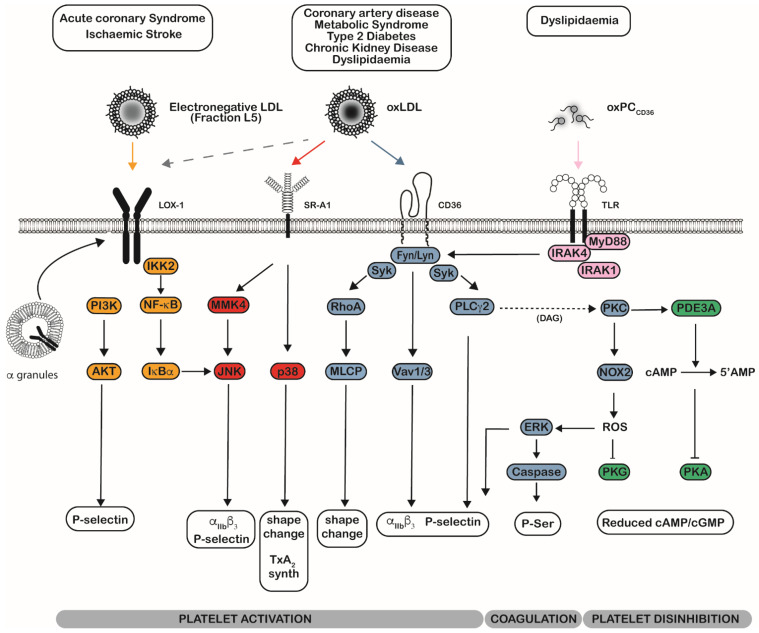
Major signalling events downstream platelet scavenger receptors induced by oxidatively modified LDL. Platelets are exposed to different types of oxidatively modified LDL, which interact with distinct pattern recognition receptors on their surface. Given that these particles have a complex and varied chemical composition, there is potential for particular LDL species to interact with different receptors. OxLDL has been shown to interact with CD36, SRA1, and potentially LOX-1. In contrast, electronegative LDL (fraction L5) has currently only been shown to interact with LOX-1. TLR2 has not been shown to bind with oxLDL but does interact with specific oxidised phospholipids that are known to be present in oxLDL (oxPC_CD36_). LOX-1 is located in the alpha granules of resting platelets and recruited to the platelet surface upon platelet activation. Once at the platelet surface, LOX-1 can bind electronegative LDL to stimulate signalling events (yellow) through PI3K and JNK to increase P-selectin expression and integrin a_IIb_b_3_ activation. SR-A1 binds oxLDL to initiate signalling through JNK and p38 (red), which can drive P-selectin expression, a_IIb_b_3_ activation, shape change, and TxA_2_. CD36 is constitutively associated with the Src family kinases Fyn and Lyn and, once ligated by oxLDL, incites a multitude of pathways (Blue). These drive shape change through RhoA and MLCP, P-selectin expression, and a_IIb_b_3_ activation through Vav1/3 and PLCg2. This pathway also leads to the activation of a PKC isoform, which is critical to the control of membrane phospholipids and inhibitory signalling (green). PKC induces the ROS generation through NOX2, which activates both ERK and phosphatidyl serine exposure, as well as the inhibition of PKG. PKC also activates PDE3A, which reduces cAMP concentrations and the activation of PKA. TLR2 does not bind oxLDL, but interaction with oxidised phospholipids leads to the MyD88-mediated activation of tyrosine kinase signalling, P-selectin expression, and a_IIb_b_3_ activation. Abbreviations: Phosphoinositide 3-kinase (PI3K), c-Jun N-terminal kinase (JNK), Myosin light chain phosphatase (MLCP), extracellular-signal regulated kinases (ERK), Protein kinase G (PKG), Protein kinase A (PKA), Protein kinase C (PKC), Phospholipase C gamma 2 (PLCg2), Vav1/3, NADPH Oxidase 2 (NOX2), p38.

**Table 1 ijms-23-09199-t001:** Functional effects of oxLDL and derivates on platelets. OxLDL—oxidised LDL; CuSO_4_—copper (II) sulphate; HUVEC—human umbilical vein endothelial cells; WT—wildtype; Acyl-LPA—acylated lysophosphatidic acid; FeSO_4_—iron (II) sulphate; MDA—malondialdehyde; HOCL—hypochlorus acid; Lyso-PC—lysophosphatidylcholine; MPO—myeloperoxidase; oxPC_CD36_—oxidised phosphocholine specific for CD36; ADP—adenosine diphosphate; METS-LDL—metabolic syndrome-associated LDL; DMII-LDL—diabetes mellitus II-associated LDL; SIN-1-LDL—3-(N-morpholino)sydnonimine-LDL; Ac-LDL—acetylated LDL; PRP—platelet-rich plasma; CuCL_2_—copper(II) chloride; Ca^2+^—calcium; NO—nitrix oxide; Cu^2+^—copper; AAPH—2,2′-Azobis-(2-amidinopropane hydrochloride).

	Environment	Functional Response	Method of Oxidation	Species	Platelet Preparation	Reference
**Adhesion**	In vitro	OxLDL induces increased platelet adhesion to collagen and HUVECs under flow/static conditions	CuSO_4_	Human	Whole blood/washed platelets	[34,40,41,42]
	In vitro	OxLDL induces increased platelet adhesion to fibrinogen under flow conditions	Electronegative LDL—fraction L5, isolated from patients	Human	Washed platelets	[33]
	In vivo	OxLDL injection increases platelet endothelial adhesion after carotid ligation in WT animals	CuSO_4_	Mouse	Whole blood	[34]
	In vivo	OxLDL injection increases platelet endothelial adhesions in cremaster muscle blood vessels	CuSO_4_	Hamster	Whole blood	[43]
	In vitro	OxLDL increases platelet spreading and pseudopodia formation under static conditions	CuSO_4/_FeSO_4_	Human	Washed platelets	[44,45,46,47,48,49,50,51,52]
	In vitro	OxLDL increases adhesion of platelets to oxLDL co-coated collagen matrix under arterial flow conditions	CuSO_4_	Human	Whole blood	[44]
	In vitro	Mildly/extensively oxidised LDL causes shape changes in platelet suspensions	O_2_ oxidation/CuSO_4_/Acyl-LPA	Human	Washed platelets	[46,53,54]
	In vitro	Increased platelet adhesion to MDA/HOCL modified-oxLDL under static conditions	MDA modified oxLDL/HOCL-LDL	Human	Washed platelets	[55]
**Degranulation & Secretion**	In vitro	OxLDL increases platelet P-selectin expression	CuSO_4_/LysoPC	Human/cat	Whole blood/washed platelets	[34,56,57,58,59]
	In vitro	Positive correlation of platelet P-selectin, sCD40L and oxLDL levels	MPO-mediated LDL oxidation	Human	Washed platelets	[60]
	In vitro	OxPC_CD36_ induces increased platelet P-selectin expression	OxPC_CD36_	Human	Washed platelets	[61]
	In vitro	OxPC_CD36_ induces P-selectin expression	KODA-PC	Mouse	Washed platelets	[62]
	In vitro	OxLDL potentiates ADP-induced P-selectin expression in platelets	HOCL-LDL	Human	Washed platelets	[63]
	In vitro	OxLDL induces platelet serotonin secretion	CuSO_4_	Human	Washed platelets	[64,65]
	In vitro	OxLDL potentiates ADP-induced serotonin secretion in platelets	HOCL–LDL	Human	Washed platelets	[63]
	In vitro	Binding of oxLDL on dyslipidaemic platelets linked to increased P-selectin expression	oxLDL-associated dyslipidaemia	Human	Washed platelets/whole blood	[34,66,67]
	In vitro	OxLDL potentiates the effect of thrombin on platelet P-selectin expression	CuSO_4_	Human	Washed platelets	[68]
	In vitro	OxLDL induces platelet CD147 release	CuSO_4_	Human	Washed platelets	[69]
	In vitro	OxLDL induces platelet thromboxane A2 generation	CuSO_4_	Human	Washed platelets	[70,71]
	In vitro	OxLDL induces platelet thromboxane B2 generation	METS-LDL/DMII LDL/glycooxidised LDL	Human	Washed platelets	[28,30]
	In vitro	OxLDL induces platelet CXCL12 release	CuSO_4_	Human	Washed platelets	[72]
**Aggregation**	In vitro	OxLDL induces platelet aggregation	CuSO_4_/Ac-LDL/HOCL-LDL/SIN-1-LDL/Acyl-LPA/O_2_ oxidation/Electronegative LDL—Fraction L5/METS-LDL/DMII-LDL/	Human	Washed platelets/ PRP	[28,46,54,73,74,75,76,77,78]
	In vitro	OxLDL potentiates the effects of ADP on platelet aggregation	CuSO_4_/HOCL-LDL/Electronegative LDL/oxLDL-associated APOE^−/−^ dyslipidaemia	Human/mouse	Washed plt./PRP/PRP and washed platelet in ratio 1:1	[33,63,74,79,80,81]
	In vitro	OxLDL potentiates the effects of thrombin aggregation	CuSO_4_/HOCL-LDL	Human	Washed platelets	[74,82]
	In vitro	OxPC_CD36_ induces increased fibrinogen binding in platelets	OxPC_CD36_	Human	Washed platelets	[61]
	In vitro	OxLDL potentiates the effects of epinephrine and thrombin in aggregation	HOCL-LDL	Human	Washed platelets	[63]
	In vitro	oxLDL increases active conformation of a_IIb_ß_3_	CuSO_4_	Human/mouse	Washed platelets	[42,62]
**Ca^2+^** **mobilisation**	In vitro	OxLDL induces intracellular Ca^2+^ mobilisation	CuSO_4_/HOCL-LDL	Human	Washed platelets	[63,68,83]
**Disinhibition**	In vitro	OxLDL inhibits activation of guanylate cyclase	CuCl_2_	Bovine	Washed platelets	
	In vitro	OxLDL decreases platelets sensitivity to NO in thrombin aggregation	CuSO_4/_O_2_ Oxidation	Human	Washed platelets	[84,85,86]
	In vitro	OxLDL reduces the effect of endothelial cell-derived NO on platelets	Not defined	Human	Washed platelets	[87]
	In vivo/In vitro	oxLDL, oxPC_CD36_ and oxPC_CD36_-associated highfat diet induces PDE3A and leads to reduced cAMP signalling	CuSO_4/_oxPC_CD36_/High fat diet	Human/mouse	Washed platelet, PRP, whole blood	[88]
	In vivo/In vitro	oxLDL-induced ROS production inhibits the effects of 8pCPT-cGMP	CuSO_4_	Human/mouse	Whole blood	[89]
**ROS-generation**	In vitro	oxLDL induces ROS generation	CuSO_4_	Human	Washed platelets	[42,89,90]
**Cell-Cell-interactions**	In vitro/In vivo	OxLDL increases platelet-leucocyte interaction and platelet-induced leucocyte transmigration	HOCL/CuSO_4_	Human/mouse	PRP/leucocyte enriched PRP	[91,92]
	In vivo	OxLDL increases platelet-leucocyte interactions and P-selectin-dependent leucocyte endothelial interactions	Cu^2+^	Hamster	Whole blood	[93]
	In vitro	OxLDL Increases platelet-monocyte interactions	Acyl-LPA	Human	Whole blood	[54]
	In vitro	OxLDL treated platelets induce ICAM-1 expression in endothelial cells	HOCL	Human	Washed platelets	[56]
**Miscellaneous**	In vivo	Injection of OxLDL shortens tailbleeding time	Electronegative LDL, fraction L5, isolated from patients	Mouse	Whole blood	[33]
	In vitro	Oxidised LDL attenuates Fibrinogen binding	FeSO_4_ (>30% assessed by REM)	Human	Washed platelets	[94]
	In vitro	OxLDL increases CXC5 release in platelets from CAD patients	CuSO_4_	Human	PRP	[95]
	In vitro	OxLDL induces microvesicle release and increases platelet phosphatidylserine exposure	CuSO_4_	Human	Washed platelets	[96]
	In vitro	OxLDL increases platelet prothrombinase activity	CuSO_4_ + AAPH/MPO oxidised LDL	Human	Washed platelets	[97]
	In vivo	oxLDL injection increases time to occlusion in a ferric chloride thrombosis model carotid artery	CuSO_4_	Mouse	Whole blood	[42]

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
