# Peer review of "Oxidised Low-Density Lipoprotein-Induced Platelet Hyperactivity—Receptors and Signalling Mechanisms"

_ijms, 2022, doi:10.3390/ijms23169199_

Round 1
Reviewer 1 Report
ijms-1823497-Revision
Berger and Naseem aim with this review to summarise recent advances of the pro-thrombotic signalling events induced in platelets by modified LDL ligation, describing the contribution of individual platelet scavenger receptors and highlighting the potential future challenges of targeting these pathways.
The ability to understand and modulate mechanisms governing interactions between LDL-cholesterol and platelets may offer new treatment strategies for atherosclerosis prevention.
Given that the mechanisms underlying platelet activation in atherosclerosis are still incompletely understood, several major points must be define for pubblication.
1. The role of platelet-derived extracellular vesicle which disseminate the procoagulant and proinflammatory effects of activated platelets, thus contributing to platelet interactions with other cells and with ox-LDL particles, must be described.
2. The interactions between platelets and ox-LDL affect several cell population, including inflammatory cells, endothelial cells and foam cell generation, all of which contribute to atherosclerosis. These interaction must be insert.
3. A more attracting and clear figure coud be provide.
ijms-1823497-Revision
Berger and Naseem aim with this review to summarise recent advances of the pro-thrombotic signalling events induced in platelets by modified LDL ligation, describing the contribution of individual platelet scavenger receptors and highlighting the potential future challenges of targeting these pathways.
The ability to understand and modulate mechanisms governing interactions between LDL-cholesterol and platelets may offer new treatment strategies for atherosclerosis prevention.
Given that the mechanisms underlying platelet activation in atherosclerosis are still incompletely understood, several major points must be define for pubblication.
1. The role of platelet-derived extracellular vesicle which disseminate the procoagulant and proinflammatory effects of activated platelets, thus contributing to platelet interactions with other cells and with ox-LDL particles, must be described.
2. The interactions between platelets and ox-LDL affect several cell population, including inflammatory cells, endothelial cells and foam cell generation, all of which contribute to atherosclerosis. These interaction must be insert.
3. A more attracting and clear figure coud be provide.
ijms-1823497-Revision
Berger and Naseem aim with this review to summarise recent advances of the pro-thrombotic signalling events induced in platelets by modified LDL ligation, describing the contribution of individual platelet scavenger receptors and highlighting the potential future challenges of targeting these pathways.
The ability to understand and modulate mechanisms governing interactions between LDL-cholesterol and platelets may offer new treatment strategies for atherosclerosis prevention.
Given that the mechanisms underlying platelet activation in atherosclerosis are still incompletely understood, several major points must be define for pubblication.
1. The role of platelet-derived extracellular vesicle which disseminate the procoagulant and proinflammatory effects of activated platelets, thus contributing to platelet interactions with other cells and with ox-LDL particles, must be described.
2. The interactions between platelets and ox-LDL affect several cell population, including inflammatory cells, endothelial cells and foam cell generation, all of which contribute to atherosclerosis. These interaction must be insert.
3. A more attracting and clear figure coud be provide.

Author Response
We would like to thank the reviewer for their comments. Please find our reply attached.

Reviewer 2 Report
The Manuscript: „ Oxidised low density lipoprotein induced platelet hyperactivity – receptors and signalling mechanism’’ by Berger and Naseem summarises recent advances in the understanding of pro-thrombotic signalling events induced in platelets by modified LDL ligation and describes the contribution of individual platelet scavenger receptors. OxLDL generated in the hyperlipidemic state are generally known to contribute to unregulated platelet activation during thrombosis and stimulate platelet activation through phosphorylation of the regulatory light chains of the contractile protein myosin IIa (MLC). OxLDL are also reported to be involved in various diseases, including cardiovascular diseases and atherogenesis.
The submitted manuscript nicely reviews ample literatures addressing the biochemical mechanisms translating atherogenic lipid stress in the form of oxLDL. After going through the manuscript, I have a couple of comments for the authors:
1. Oxidative stress is known to be an important risk factor for the pathogenesis of various diseases. Please discuss with relevant reference(s) how oxidation of LDL contributes in pathogenesis of different diseases.
2. OxLDL is also used to measure the protein damage due to the oxidative modification of the ApoB subunit on LDL cholesterol. Please briefly discuss this point in the manuscript.
Author Response

(The authors gave the same response as above.)

Round 2
Reviewer 1 Report
The role of the reviewer is improve the scientific interest of the manuscript/review.
Any considerations/suggestions of reviewer has been included in your revision.